# Effects of Nitrogen Fertilizer on Quality Characteristics of Wheat with the Absence of Different Individual High-Molecular-Weight Glutenin Subunits (HMW-GSs)

**DOI:** 10.3390/ijms23042178

**Published:** 2022-02-16

**Authors:** Tao Yang, Qin Zhou, Qi Wang, Xiao Wang, Jian Cai, Mei Huang, Dong Jiang

**Affiliations:** College of Agriculture, Nanjing Agricultural University, No. 1 Weigang Road, Nanjing 210095, China; 2017101046@njau.edu.cn (T.Y.); 2016101052@njau.edu.cn (Q.W.); xiaowang@njau.edu.cn (X.W.); jiancai@njau.du.cn (J.C.); meihuang@njau.edu.cn (M.H.)

**Keywords:** wheat, HMW-GSs, absence, nitrogen fertilizer, gluten, gene expression, nitrogen metabolism enzyme

## Abstract

High-molecular-weight glutenin subunits (HMW-GSs) are important components of gluten, which determine the grain quality of wheat. In this study, we investigated the effects of nitrogen (N) fertilizer application on the synthesis and accumulation of grain protein and gluten quality in wheat lines with different HMW-GSs absent. The results showed that the absence of the HMW-GS in the wheat variety Ningmai 9 significantly decreased the contents of gluten, glutenin macropolymer (GMP), protein compositions, HMW-GS and HMW-GS/LMW-GS. The reduction in glutenins was compensated to some extent by an increase of gliadins. The absence of x-type HMW-GSs (1, 7 and 2 subunits) had a greater effect on gluten and GMP properties than y-type HMW-GSs (8 and 12 subunits). The content of protein compositions, gluten and GMP increased with an increase of N level; however, the increment in wheat lines with the absence of HMW-GS, especially in Ax1a, Bx7a and Dx2a, was lower than that in the wild type under various N levels. The expression level of genes encoding HMW-GSs, and activities of nitrate reductase (NR) and glutamine synthetase (GS), differed significantly among the investigated wheat lines. The reduction in gene expression and activities in Ax1a and Dx2a may account for the reductions in gluten, GMP, protein compositions, HMW-GS and HMW-GS/LMW-GS.

## 1. Introduction

Wheat (*Triticum aestivum* L.) is a staple food crop for humans in the world [1]. The unique elasticity and extensibility of dough enables wheat flour to be processed into a wide range of foods, such as bread, biscuit and noodles. The elasticity and extensibility are conferred by gluten proteins classified as gliadins and glutenins. The gliadins are monomeric proteins, whereas glutenins are polymeric proteins and are further divided into high-molecular-weight subunits (HMW-GSs, 65–90+ KDa) and low-molecular-weight subunits (LMW-GSs, 30–60 KDa) [2]. Although HMW-GSs account for ~10% of grain protein, they are major determinants of gluten elasticity and functionality [3].

Variation in HMW-GSs is controlled by alleles at the Glu-A1, Glu-B1 and Glu-D1 locus on the long arms of chromosomes 1A, 1B and 1D, respectively [4]. HMW-GSs are coded by Glu-1 loci and divided into a higher molecular weight x-type subunit and a lower molecular weight y-type subunit. Both x- and y-type HMW-GSs have a typical three-domain structure consisting of relatively small N- and C-terminal domains and a major central domain. Common wheat usually has five different HMW-GSs. Gene silencing is an effective approach to investigate the function of HMW-GSs in food processing. Song et al. [5] reported that the absence of the Dx2 subunit weakened the gluten network and dough properties. Meanwhile, Liu et al. [6] reported that the absence of the Ax1 subunit decreased the protein content and formed larger apertures in the microstructure of gluten. Zhang et al. [7] suggested that the absence of the HMW-GS in Glu-D1 showed much lower gluten and dough strength than in Glu-A1 and wild genotype. Grain protein synthesis is largely dependent on nitrogen assimilation and protein hydrolysis governed by several enzymes in the vegetative organs of wheat. Enzymes of nitrogen metabolism play important roles in modulating protein synthesis in wheat grain and the export of amino acids [8]. Therefore, it is of interest to study the relationships between the variation of HMW-GSs and gluten synthesis.

The quality of wheat is affected by the genotype, environment, cultivation management and their interactions [9]. Nitrogen fertilizer is a main factor in increasing yield and regulating grain protein content and quality in wheat [10]. Zheng et al. [11] reported that reducing N rate improved grain quality and nitrogen agronomic efficiency of soft wheat but resulted in grain yield loss. The application of N fertilizer in the jointing stage significantly increased the ratios of glutenin/gliadin and of HMW/LMW-GS [12]. Ma et al. [13] reported that the split of nitrogen fertilizer increased grain yield and soil inorganic N supply in the later growth stages of wheat. Our previous research revealed that, compared to other development stages of wheat, N topdressing at the top-first-leaf stage greatly improved grain yield and gluten protein content [14]. Meanwhile, other studies have investigated the effect of N application on the gluten property in wheat with variations in HMW-GSs. Song et al. [5] reported that N treatment at the booting stage was highly effective in improving the gluten structural and thermal properties of wheat, especially of glutenin subunits (Dx2 + Dy12). Daaloul Bouacha et al. [15] reported that N application appeared to enhance the expression of y-type HMW-GS rather than x-type HMW-GS.

As the main raw material for biscuit and cake making, the food industry’s demand for high-quality soft wheat grains is increasing. In particular, wheat grains with a low content of protein and gluten are required to achieve a good biscuit-making quality. However, in China, a large gap exists between the quality and yield of soft wheat grain and requirements for end-use quality. Nitrogen fertilizer application strategies are effectively used to regulate the yield of soft wheat; however, the protein and gluten content are promoted simultaneously, which may reduce biscuit-making quality. The absence of HMW-GSs in wheat is negatively correlated with protein content and gluten strength, which may be beneficial for soft wheat quality. Ningmai 9 is a special soft wheat variety with wide adaptability in production. The absence of HMW-GS at Glu-A1, B1 and D1 loci in Ningmai 9 is a useful strategy to reduce the protein and gluten content in grain. Therefore, the effects of N application on the yield and grain quality in wheat lines with the absence of HMW-GS in the Ningmai 9 variety should be further explored to improve the yield and quality of soft wheat.

In this study, a group of wheat lines was used to investigate the absence of different individual HMW-GSs on the quality and functionality of gluten protein. The response to N fertilizer was explored to improve yield and grain quality. Our results provide insights into how the absence of different HMW-GSs influenced the quality of wheat, and provide a reference for the production of high-quality soft wheat with a high or stable yield.

## 2. Results

### 2.1. Analysis of Grain Yield of Wheat Lines under Different N Treatments

Grain yields increased significantly for all wheat lines with the increases in N application, and peaked at N180 level (Table 1), with the increments of 91.41%, 56.75%, 73.13%, 66.00%, 61.04% and 73.34% from N0 to N180 for WT, Ax1a, Bx7a, By8a, Dx2a and Dy12a, respectively. There was no significant difference in the 1000-kernel weight between all wheat lines except for Ax1a, Bx7a and Dx2a at different N levels. The spikes per pot of Ax1a and Dy12a were close to that of WT at N0 and N120 levels, and there was no significant difference in spike number per pot between all wheat lines except for By8a at N180 level. The grain yield of wheat lines with the absence of HMW-GS decreased compared with WT, which might be due to the decrement of kernels per spike at different N levels, where the decrements of kernels per spike for Ax1a, Bx7a, By8a, Dx2a and Dy12a were 10.48%, 12.89%, 12.29%, 7.89% and 14.87% compared with WT at N180 level, respectively.

### 2.2. Grain Quality Traits of Wheat Lines under Different N Treatments

The content of total protein, protein components and wet and dry gluten of all wheat lines significantly increased with the increase in N levels (Figure 1). The absence of HMW-GS had no significant effect on protein content (except for By8a) under different N levels (Figure 1A). The increase rate of protein content in By8a was lower than that in WT from N0 to N120 and N180. The wet gluten contents of Ax1a, Dx2a and Dy12a were lower than that in WT at different N levels, with the decrements of 2.39%, 1.67% and 4.03% for Ax1a, 8.05%, 1.67% and 3.92% for Dx2a, and 3.00%, 2.52% and 0.25% for Dy12a at N0, N120 and N180 levels, respectively (Figure 1(B-1)). The dry gluten content showed a similar change pattern under different N levels (Figure 1(B-2)). The gluten index of wheat lines with the absence of HMW-GS significantly decreased compared with WT, especially for Ax1a and Bx7a, which decreased by 15.37%, 11.03% and 12.33% for Ax1a and 21.82%, 21.43% and 22.08% for Bx7a at N0, N120 and N180 levels, respectively (Figure 1(B-3)). The absence of HMW-GS significantly decreased the contents of glutenin, albumin and globulin, while it increased gliadin contents in Ax1a, Bx7a and Dx2a compared with WT at different N levels (Figure 1(C-1) and Figure 2, Figure 3 and Figure 4). The Ax1a, Bx7a and Dx2a had a much lower glutenin content compared with WT at N120 and N180 levels, which decreased by 22.72% and 24.88% for Ax1a, 21.43% and 22.08% for Bx7a, and 21.47% and 20.75% for Dx2a at N120 and N180 levels, respectively.

### 2.3. HMW-GS and LMW-GS Contents of Wheat Lines under Different N Treatments

HMW-GSs and LMW-GSs in six wheat lines were identified, and the variation of HMW-GS composition encoded by different Glu-1 loci in six wheat lines under different N treatments are shown in Figure 2. The content of HMW-GS and LMW-GS of six wheat lines significantly increased with the increase of N level (Figure 3(A-1,A-2)). The absence of HMW-GS significantly decreased HMW-GS content under different N levels, where Ax1a, Bx7a and Dx2a had much lower HMW-GS content than that in WT, which decreased by 56.62%, 42.11% and 56.60% at N0 level, 57.93%, 40.23% and 63.48% at N120 level, 31.39%, and 23.64% and 26.49% at N180 level, respectively. The LMW-GS content in Bx7a, By8a and Dx2a was lower than that in WT under different N levels, with the decrements of 16.51%, 3.68% and 27.76% at N0 level, 15.69%, 3.49% and 23.96% at N120 level, and 20.01%, 9.09% and 26.68% at N180 level, respectively. While LMW-GS content in Ax1a and Dy12a was higher than that in WT under different N levels. As shown in Figure 3(A-3), the ratios of HMW-GS/LMW-GS in wheat lines with the absence of HMW-GS (except for By8a) decreased or fluctuated as compared with WT at N0 and N120 levels. While the ratios of HMW-GS/LMW-GS of Ax1a, Dx2a and Dy12a increased rapidly from N120 to N180 compared with WT, with the increments of increase rate of 52.86%, 94.91% and 40.72%, respectively.

As shown in Figure 3B, Bx7 and Dx2 subunits had higher proportions in total HMW-GS among all wheat lines than other individual HMW-GS. The content of different individual HMW-GS increased with the increase of N level. The content of Bx7, Dx2 and Dy12 subunits was lower in the wheat lines with the absence of HMW-GS compared with WT at different N levels. While the Ax1 subunit content in Bx7a, Dx2a and Dy12a was lower than that in WT at N0 and N120 levels, and there was no significant difference between Dx2a, Bx8a and WT at N180 level. The content of the By8 subunit was higher in Dy12a compared with WT at different N levels.

### 2.4. Free Sulfhydryl (SH), Glutenin Macropolymer (GMP) Content and Dynamic Rheological Properties of GMP Gels of Wheat Lines under Different N Treatments

Free SH content ranged from 0.782 to 0.884 μmol/g in all wheat lines, and increased significantly with the increase of N application (Table 2). The absence of HMW-GS significantly decreased free SH content at N0 level (except for By8a) and N180 level (except for Dy12a), and the decrements of free SH content for Ax1a, Bx7a, Dx2a and Dy12a were 7.03%, 7.71%, 10.96% and 7.71% at N0 level, and for Ax1a, Bx7a and Dx2a were 5.19%, 6.51% and 4.42% at N180 level, respectively, as compared with WT, while there was no significant difference for any wheat lines at N120 level. Moreover, the lower increase rate of free SH content in Ax1a and Dx2a was observed compared with WT (20.83%) from N120 to N180 level, with the increments of 12.50% and 13.21%, respectively.

GMP contents in wheat lines with the absence of HMW-GS decreased at different N levels compared with WT, especially in Ax1a, Bx7a and Dx2a at N120 and N180 levels (Table 2), with the decrements of 25.57%, 23.92% and 19.26% at N120 level, and 25.41%, 22.22% and 17.47% at N180 level, respectively. The GMP content and wet GMP-gel weight of different wheat lines increased significantly with increased N level. While the increase rates of GMP contents in Ax1a, Bx7a, Dx2a and Dy12a were 2.11, 1.93, 1.86 and 1.93 times lower than that of the WT from N0 to N120 level, and 1.83, 1.53, 1.49 and 2.03 times lower than that of the WT from N0 to N180 level, respectively. The dynamic rheological properties of GMP gels were determined to investigate the viscoelasticity of GMP gels. The elastic modulus (G′) in the wheat lines with the absence of HMW-GS decreased except for that in By8a and Dy12a, while viscous modulus (G″) increased except for that in Dy2a compared with WT at different N levels. The N application greatly increased the G′ and G″ for all wheat lines. The value of loss angle (δ) can comprehensively reflect the properties of elastic and viscous modulus. The δ value decreased largely with increased N level for all wheat lines and increased in the wheat lines with the absence of HMW-GS compared with WT except for Dy12a. Ax1a had the highest δ value at different N levels, which increased by 37.06%, 30.72% and 33.72%, respectively, as compared with WT under different N levels.

### 2.5. Dynamic Changes in Total Grain Protein of Wheat Lines during Grain Development under Different N Treatments

Dynamic changes of wheat grain protein during grain filling are shown in Figure 4. The grain protein content showed a concave curve which first decreased from 10 days after anthesis (DAA) to 20 DAA and then increased from 20 DAA to the mature stage in all wheat lines. The N application increased the grain protein content during grain filling for all wheat lines. The absence of HMW-GS influenced the dynamic changes of grain protein content during grain filling at different N levels. Compared with WT, the grain protein content in Ax1a and Bx7a decreased at 10 DAA under N0 level, while it showed an inverse trend at 20 DAA and then kept a lower content in Ax1a and Dx2a at the mature stage. At N120 level, the grain protein content in Ax1a and Dx2a decreased at 10 DAA compared with WT, while the grain protein content among all wheat lines was close to 20 DAA and the mature stage. At N180 level, the grain protein content in Dx2a decreased compared with WT over the whole grain development.

### 2.6. Changes in Nitrate Reductase (NR) and Glutamine Synthetase (GS) Activities in Flag Leaves of Wheat Lines during Grain Development under Different N Treatments

The NR and GS activities in the flag leaves exhibited similar patterns during grain-filling for all wheat lines at different N levels (Figure 5). The activities of the two enzymes both decreased from flowering stage to 20 DAA, and significantly increased with increase of N level in all wheat lines. The absence of HMW-GS influenced the two enzymes activities during grain-filling. The NR activities of Ax1a and Dx2a were much lower than that in WT, especially at 20 DAA under N180 level, with the decrements of 5.75% and 3.69%, respectively. While the NR activity in By8a was higher than that in WT, and the increment was 6.55%, 6.41% and 3.91% at N0, N120 and N180 levels at 20 DAA, respectively. The GS activities of all wheat lines followed similar change patterns with the NR activities; Ax1a and Dx2a had lower GS activities than those in WT at N180 level, which decreased by 0.88%, 2.29% and 1.80% for Ax1, and 3.76%, 2.53% and 0.79% for Dx2 at flowering stage, 10 DAA and 20 DAA, respectively. In contrast, By8a had a higher GS activity than that in WT, which increased by 3.75%, 2.53% and 0.79% at flowering stage, 10 DAA and 20 DAA, respectively.

### 2.7. Relative Expression of Genes Related to HMW-GS of Wheat Lines under Different N Treatments

Relative expression levels of HMW-GS-related genes in seeds at 20 DAA are shown in Figure 6. The relative expression patterns of these genes differed in different wheat lines with the absence of HMW-GS compared with WT. Compared with WT, the relative expression level of the *Ax1* gene was significantly lower in Dx2a, the *Bx7* gene was significantly lower in By8a, the *By8* gene was significantly lower in Bx7a and Dx2a, and the *Dx2* gene was significantly lower in Bx7a and By8a under different N levels. The relative expression levels of genes *Ax1* and *Bx7* were elevated in Dx2a and Dy12a from N0 to N180 level, with the increments of 18.64% and 27.08% for the *Ax1* gene in Dx2a, and 7.19% for the *Bx7* gene in Dy12a, respectively.

## 3. Discussion

The concentration and composition of glutenin especially the types and numbers of HMW-GSs determine the unique viscoelasticity of wheat dough [16]. The functionality and characteristics of gluten, and the synthesis and accumulation of protein were also affected by the absence of HMW-GS. Superior combinations of HMW-GSs accelerate the polymerization of glutenin during grain development, thereby increasing the gluten and GMP content [6,17,18]. However, in agreement with Gao et al. [19], the absence of HMW-GS decreased the glutenin and protein polymerization (Figure 4). To further investigate how the absence of different individual HMW-GS in Ningmai 9 affected the gluten properties and the polymerization of gluten protein, we analyzed the gluten protein and GMP properties of a group of wheat lines with the absence of HMW-GS in Ningmai 9, as well as the nitrogen metabolism levels, and the expression levels of genes related to HMW-GS during different developmental stages.

The wet gluten and GMP contents are important indicators for measuring the quality of wheat grain. In this study, the absence of HMW-GS significantly decreased wet gluten and GMP content under different N levels. We found that the wet gluten content in Bx7a and Dx2a decreased more than By8a and Dy12a compared with WT. Meanwhile, the elastic modulus (G′) in Ax1a, Bx7a and Dx2a decreased compared with WT, while loss angle (δ) values increased, which indicated that the absence of x-type HMW-GSs affected gluten and GMP qualities more than y-type HMW-GSs. These findings agree of those of H. Wieser and Zimmermann [20]. The cysteine residues existing in HMW-GS and LMW-GS are involved in inter- and intra-molecular disulfide bonding in the formation of GMPs and play an important role in the functioning of HMW-GS. The SH usually contributes to the formation of covalent bonds and cross-linking in gluten [21]. Our results showed that Ax1a and Dx2a had a lower SH content compared with WT, which was probably attributed to the lower cysteine content in Ax1a and Dx2a [2]. Meanwhile, the increased rate of SH in Ax1a and Dx2a was relatively minor compared with WT in N120 to N180, which suggested that Ax1a and Dx2a were not sensitive in response to higher N fertilizer. Moreover, the wet gluten and GMP contents increased with increased N levels, while N had less effect on the wet gluten and GMP contents of wheat lines with the absence of HMW-GS, especially Ax1a and Dx2a. This finding was consistent with Luo et al. [22] who suggested that genotypes with high stability have weak interactions with the environment.

Notably, protein contents of all wheat lines increased with increased N levels, and the differences of protein contents between wheat lines with HMW-GSs absence and WT were not significant at different N levels (Figure 1A). However, the absence of HMW-GS decreased the contents of glutenin, albumin and globulin, while increased gliadin contents compared to the WT under different N levels (Figure 1C), indicating that the reduction in glutenins was compensated to some extent by the increase in gliadins [23]. This was another reason that decreased the gluten and GMP content in wheat lines with the absence of HMW-GS, because gliadins could weaken the interactions between glutenin chains and essentially have a plasticizing effect on the gluten structure and further impact the gluten and GMP properties [24]. In this study, the absence of Ax1, Bx7 and Dx2 subunits significantly decreased the HMW-GS content and the ratio of HMW-GS/LMW-GS at N0 and N120 levels. As the major skeleton of GMP, HMW-GSs contribute to formation of GMP via intra-/intermolecular disulfide bonds and hydrogen bonds and are major determinants of gluten structure and behavior [25]. Therefore, we speculated that the decreased HMW-GS/LMW-GS ratio and increased gliadin content in Ax1a, Bx7a and Dx2a may be related to the inefficient formation of GMP skeleton. The free SH content showed that the absence of HMW-GS led to a decrease in free SH group level, which agrees with the above results. Compared with WT, the relative expression levels of the *By8* and *Dx2* genes were down-regulated in Bx7a, and the *Ax1* and *By8* genes were down-regulated in Dx2a under different N levels, which was consistent with the results of individual HMW-GS changes in Bx7a and Dx2a. Meanwhile, the *Ax1* and *Dx2* genes were down-regulated in Dx2a and Bx7a, respectively. Correspondingly, the content of individual HMW-GSs also decreased in wheat lines. It showed that the down-regulation in expression of individual gene encoding HMW-GSs led to the reduction in HMW-GS content. Therefore, the lower gene expression may also one of the reasons for lower contents of wet gluten, GMP, glutenin and HMW-GS in Bx7a and Dx2a.

Since N has a greater effect on the grain yield and gluten content than genetic factors [26], increasing N rates is an effective practice for increasing both yield and gluten content and its compositions in wheat. However, Ningmai 9 is a soft wheat variety, which is usually used for biscuit and cookies. The increase in gluten content after N application will reduce the baking quality of Ningmai 9. In this study, the content of protein and its composition, gluten, GMP, free SH, HMW-GS and LMW-GS of all wheat lines significantly increased with the increased N levels. However, the content of glutenin, HMW-GS and LMW-GS in the wheat lines with the absence of HMW-GS, especially in Bx7a and Dx2a, were lower compared to the WT. These results indicated that wheat lines with the absence of HMW-GS were not sensitive in response to the increase of N with regard to protein components. N application significantly increased the yield of all wheat lines, meanwhile, the absence of HMW-GS significantly decreased the gluten and GMP content compared with the WT, which is beneficial for soft wheat quality. Therefore, the wheat lines with the absence of HMW-GS, especially the Bx7a and Dx2a, can be used to coordinate yield and quality.

The accumulation of grain protein in wheat generally depends on the synthesis of amino acids and nitrogen metabolism [27]. The activity of GS and NR are strongly correlated with nitrogen metabolism [28]. Li et al. [29] reported that higher GS and NR activities in the flag leaves can increase the total protein and its composition. In this study, Ax1a and Dx2a had lower NR and GS during grain filling compared with WT, especially at 20 DAA under N180 level. It indicated that the absence of Ax1 or Dx2 delayed the assimilation of nitrogen at higher N level, and inhibited the ability of nitrogen accumulation and use and the transport of nitrogen from leaf to grain, thereby leading to the lower availability of amino acids for subsequent protein synthesis. The results of dynamic changes of protein content in wheat grain during grain-filling showed that, compared with WT, the accumulation of protein in Ax1a, Bx7a and Dx2a was relatively lower at different N levels, especially the lower protein accumulation in Dx2a under N180 level at the middle and late grain-filling stages (Figure 4). These findings confirmed the above results that the lower response of protein synthesis in Dx2a is due to N fertilizer application. Collectively, these results suggested that the amino acids were available for the subsequent synthesis of the grain proteins in Ax1a and Dx2a compared with WT, which may explain the lower contents of glutenin, HMW-GS and LMW-GS in Ax1a and Dx2a.

## 4. Materials and Methods

### 4.1. Experimental Design

A semi-field experiment was carried out in Tangquan Experiment Station of Nanjing Agriculture University, Nanjing (118°27′ E, 32°05′ N), Jiangsu Province, China in the wheat growing seasons of 2016–2017 and 2017–2018. The average temperature in the growing season are shown in Appendix A, the average precipitations were 105 mm and 83 mm, the average temperature were 12.41 °C and 11.74 °C in 2016–2017 and 2017–2018 growing seasons, respectively. The shelters were used to prevent natural precipitation throughout the growth stage of wheat plants. A group of different individual HMW-GS-absent lines of Ningmai 9 were used in this study, which was obtained from Jiangsu Academy of Agricultural Sciences, Nanjing, Jiangsu province, where the wheat lines were obtained using ethyl methane sulfonate (EMS) mutation. The HMW-GS compositions of wild type of Ningmai 9 contains subunits 1, 7 and 2 as x-type subunits, and subunits 8 and 12 as y-type subunits. Five wheat lines were absent in the HMW-GS of Ax1, Bx7, By8, Dx2, and Dy12, respectively, which was designated as Ax1a, Bx7a, By8a, Dxa2 and Dy12a, and the wide type was designated as WT. The six wheat lines (5 deletion lines and 1 WT) were planted with three N levels, including 0 kg N ha^−1^ (N0, 0 g N/pot), 120 kg N ha^−1^ (N120, 0.629 g N/pot) and 180 kg N ha^−1^ (N180, 0.943 g N/pot). The experiment was randomly designed with three replicates. Overall, 540 pots were prepared at each growing year.

In total, 21 wheat seeds were planted in one pot and thinned to seven seedlings at third leaf stage. Each pot (22 cm in height and 25 cm in diameter) was filled with 7.5 kg of soil. N was applied as basal and topdressing fertilizer with the ratio of 7:3, and topdressing fertilizer was applied at jointing stage. In total, 1 g KH_2_PO_4_ was applied as a basal fertilizer. Uniform plants flowering on the same day were tagged for sampling. The labelled flag leaves and grains were sampled at flowering stage, 10 DAA and 20 DAA. About half of the samples were stored in −80 °C to analyze the enzyme activities and gene expression levels. The other samples were used to determine the protein content. At maturity, grains were harvested and cleaned. After 2 months of storage, grains were milled into flour using miller (ZS70-II, grain and oil foodstuff machine factory, Zhuozhou, China) with a 100 μm mesh sieve.

### 4.2. Determination of Yield and Yield Components

At maturity, all plants in one pot were harvested to determine spike number, grain number (GN) per spike, 1000-grain weight (TGW) and yield.

### 4.3. Determination of Grain Quality Traits

The contents of total protein and of the protein components (albumin, globulin, glutenin and gliadin) were determined by the micro-Kjeldahl method AACC (2000) [30] with coefficient of 5.7. The protein components were sequentially extracted four times from 1 g flour with distilled water, 10% NaC1, 75% ethanol and 0.2% NaOH, respectively. After extraction, the collected supernatants were dried for measurement of N by the micro-Kjeldahl method. The contents of wet gluten, dry gluten and gluten index were determined according to the AACC 38-12.02 procedure [30] with a gluten instrument (Perten instruments AB, Stockholm, Sweden). Total HMW-GS and LMW-GS were separated by our previous method [31]. The separated components were used for SDS-PAGE. Quantifications of HMW-GS and LMW-GS were conducted by software QUANTITY ONE following our previous established method [32]. The free sulfhydryl (SH) content was determined according to the method of Lambrecht et al. [33]. The glutein macropolymer (GMP) content was determined by the method described by Weegels et al. [34]. The isolation of GMP was conducted according to the method reported by Wang et al. [35], and the dynamic rheological measurements of GMP gels was conducted by method described by Don et al. [36].

### 4.4. Determination of Activities of Enzymes in Flag Leaves

Nitrate reductase (NR) activity was determined according to the method of Plett et al. [37]. Glutamine synthetase (GS) activity was measured according to the method of Lillo [38].

### 4.5. Determination of Gene Expression

Total RNA Purification Kit (Genemark, GMbiolab Co., Ltd., Taichung, Taiwan) according to the manufacturer’s protocol. Approximately 500 ng purified mRNA was used to synthesize cDNA by using a Super Smart cDNASynthesis Kit (Takara Bio Inc., Shiga, Japan). The cDNA was used for quantitative real-time PCR (qRT-PCR). Relative gene expression was calculated using the comparative delta cycle threshold (Δct method) using ADP-ribosylation factor (ADP-RF, gene bank ID: 105052388) [39] as the reference gene [40]. Primers for PCR analysis were: *Ax1* gene (gene bank ID: MF568382.1), 5′-CCAGGATAATGGCAAGAACT-3′ and 5′-GAAGTTGGGTAGTATTGTGC-3′; *Bx7* gene (gene bank ID: BK006773.1), 5′-TTCGCAGCAACTCCAACAAA -3′ and 5′-GGCCTGGATAGTATGACCCC-3′; *By8* gene (gene bank ID: KY643684.1), 5′-CCTAGCTTCTCAGCAGCAGC-3′ and 5′-TTGTTTGTTGCCCTTGTCCT-3′; *Dx2* gene (gene bank ID: KF466259.1), 5′-AACCAGGACAATTGCAACAA-3′ and 5′-GACCTTGTTGCCCTTGTGCT-3′. Three technical replicates of gene expression experiment were performed.

### 4.6. Statistical Analysis

All data were subjected to one-way ANOVA using SPSS Version 10.0. Mean comparisons were performed in terms of the least significant difference (LSD), at the significance level of *p* < 0.05. Figures were generated using Origin 2018 (OriginLab, Northampton, MA, USA). All the data are presented as the mean of two years, and details of two years’ data in this study are shown in Appendix A.

## 5. Conclusions

In summary, the grain yields in the wheat lines with HWM-GS in Ningmai 9 increased with the increase of N levels, and the absence of HMW-GS significantly decreased the content of wet gluten, free SH, GMP, protein composition (except for gliadins), HMW-GS and H/L ratio compared with WT at different N levels. The absence of x-type HMW-GSs (1, 7 and 2 subunits) affected gluten and GMP qualities more than y-type HMW-GSs (8 and 12 subunits). The content of protein, gluten, GMP, SH, protein composition and subunits increased with the increase of N application. However, the content of gluten, GMP, glutenin, HMW-GS and LMW-GS in the wheat lines with the absence of HMW-GS maintained a relatively lower content at different N levels compared with WT; meanwhile, the lower increase rate of these traits was also found in the wheat lines with the absence of HMW-GS with the increase of N level. This was consistent with the down expression of genes encoding HMW-GSs and with the lower activities of NR and GS in the wheat lines with the absence of HMW-GS, especially in Ax1a and Dx2a. These findings may contribute to the adaptation of soft wheat production.

## Figures and Tables

**Figure 1 ijms-23-02178-f001:**
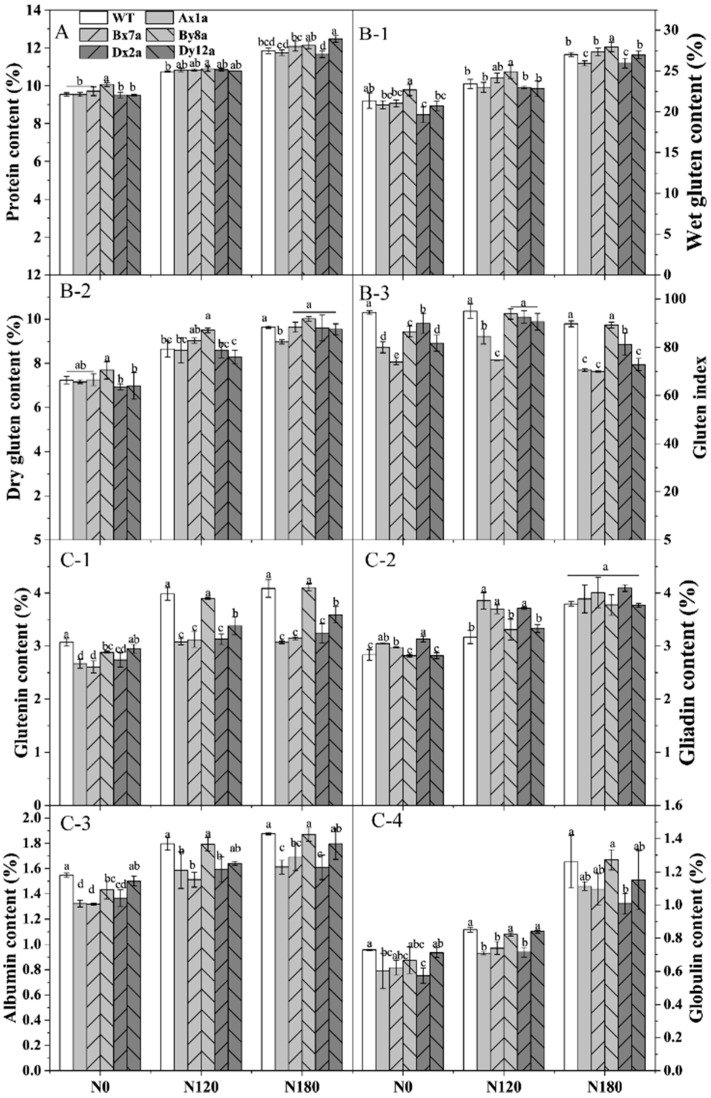
Content of protein (**A**), wet gluten (**B-1**), dry gluten (**B-2**), gluten index (**B-3**) and protein components (glutenin, **C-1**; gliadin, **C-2**; albumin, **C-3**; globulin, **C-4**) in wheat lines with the absence of HMW-GS under different N treatments. The data of protein content represent the means based on two years. Different letters in the same N level for different wheat lines indicate a significant difference (*p* < 0.05).

**Figure 2 ijms-23-02178-f002:**
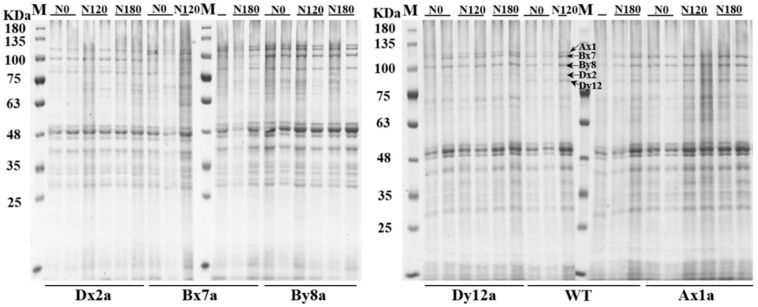
SDS-PAGE patterns of proteins in flour of wheat lines with the absence of HMW-GS under different N treatments.

**Figure 3 ijms-23-02178-f003:**
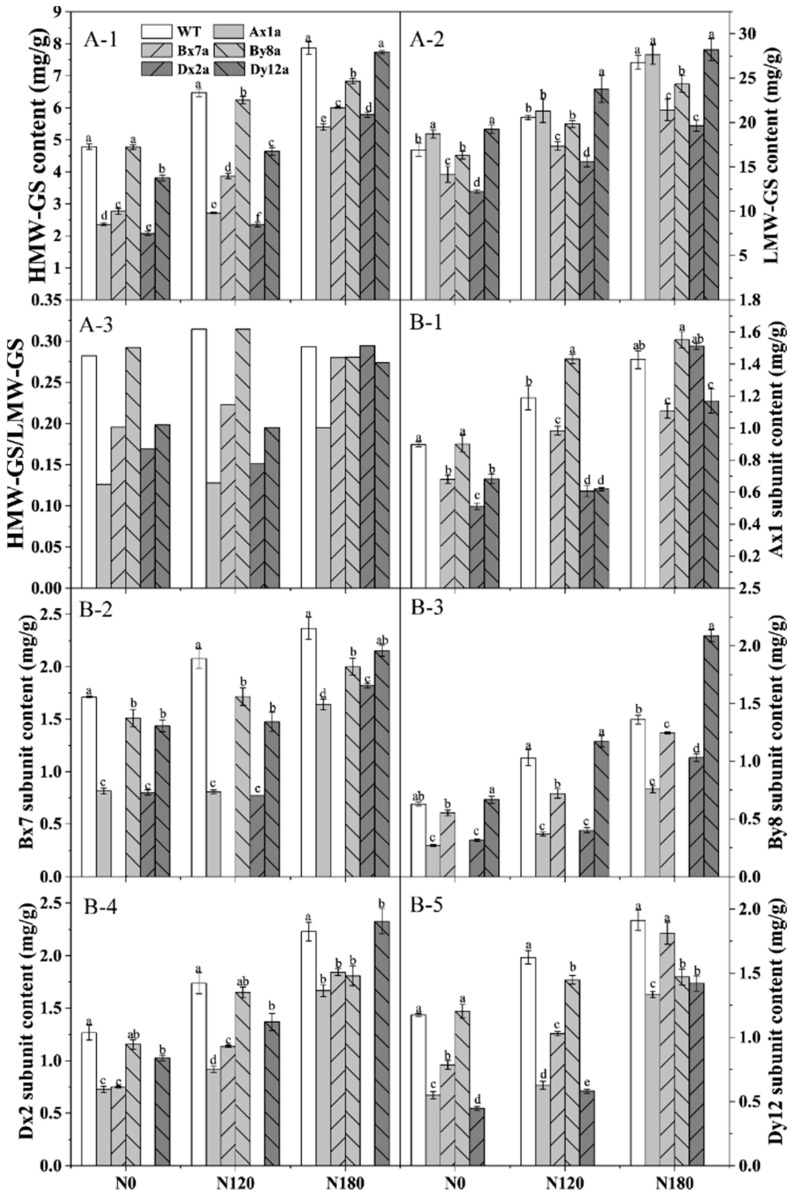
Content of HMW-GS (**A-1**), LMW-GS (**A-2**), HMW-GS/LMW-GS (**A-3**) and individual HMW-GS (**B**) (Ax1 subunit, **B-1**; Bx7 subunit, **B-2**; By8 subunit, **B-3**; Dx2 Subunit **B-4**; Dy12 subunit, **B-5**) in wheat lines with the absence of HMW-GS under different N treatments. Data represent the means based on two years. Different letters in the same N level for different wheat lines indicate a significant difference (*p* < 0.05).

**Figure 4 ijms-23-02178-f004:**
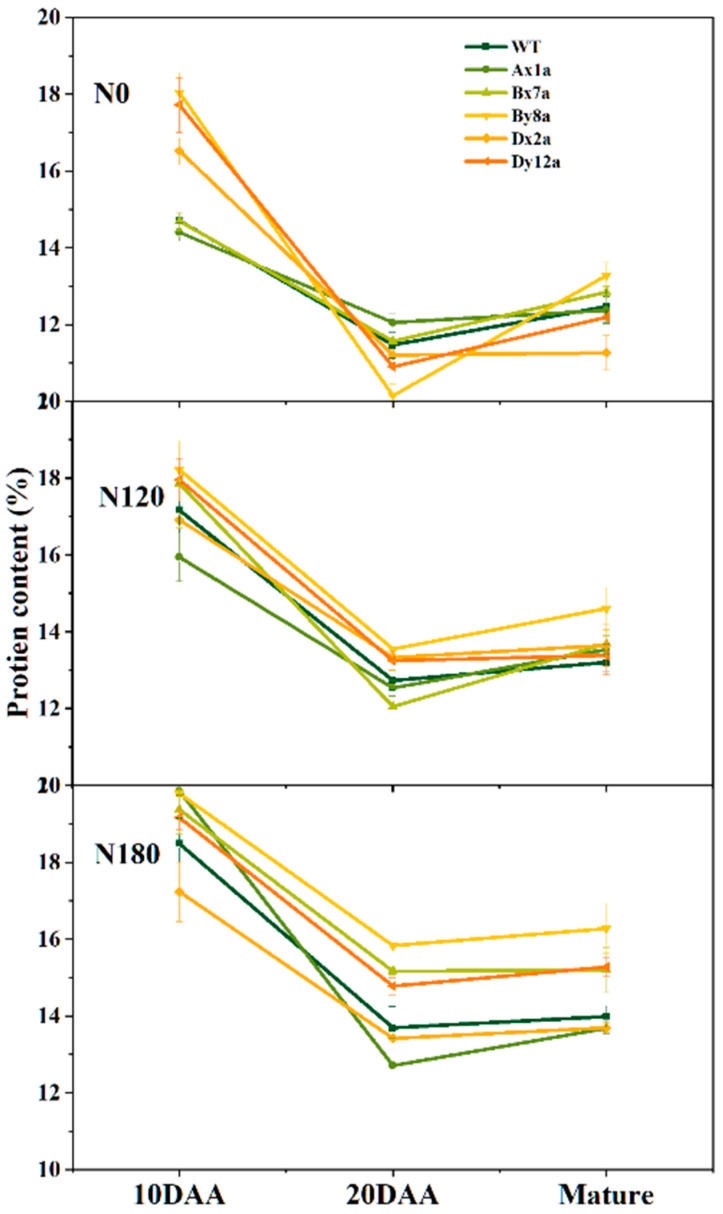
Dynamic changes in total protein content of grains in wheat lines with the absence of HMW-GS during grain development under different N treatments in 2017–2018 wheat growing season.

**Figure 5 ijms-23-02178-f005:**
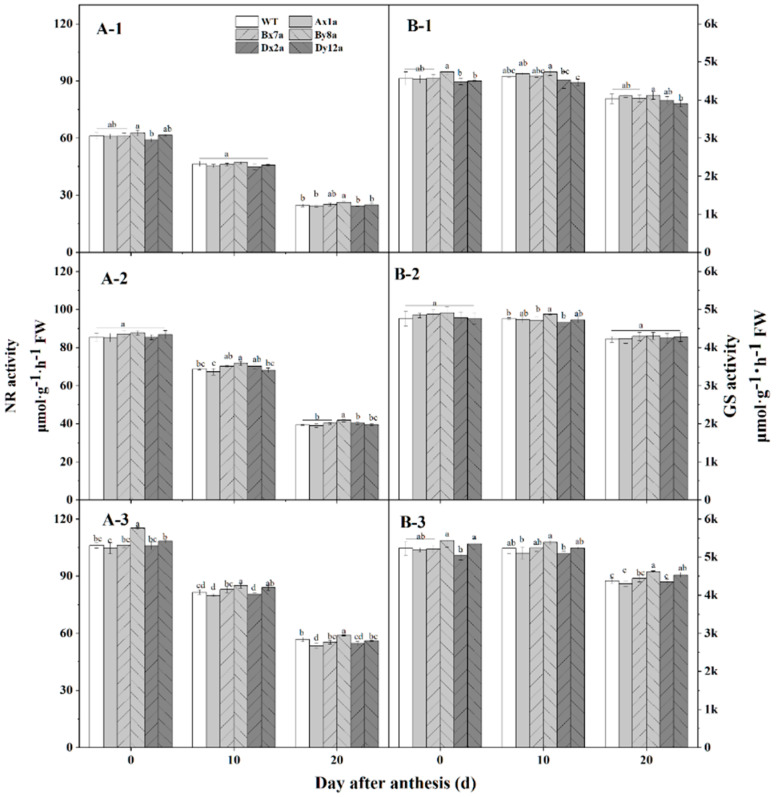
Changes in total nitrate reductase (NR) (**A**) and glutamine synthetase (GS) (**B**) activity in wheat lines with the absence of HMW-GS under different N treatments in 2017–2018 wheat growing season. N0: (**A-1**,**B-1**); N120: (**A-2**,**B-2**); N180: (**A-3**,**B-3**). Different letters in the same N level for different wheat lines indicate a significant difference (*p* < 0.05).

**Figure 6 ijms-23-02178-f006:**
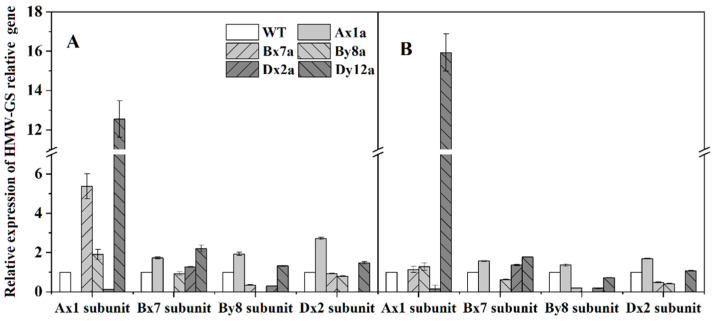
Relative expression levels of HMW-GS-related genes in grains from six wheat lines under N0 (**A**) and N180 (**B**) levels in 2017–2018 wheat growing season.

**Table 1 ijms-23-02178-t001:** Grain yield and its components of wheat lines with the absence of HMW-GS under different N treatments.

Treatment	Wheat Lines	Spikes per Pot	Kernels per Spike	1000-Kernel Weight (g)	Grain Yield per Pot (g)
N0	WT	11.67 ± 0.57 a	49.67 ± 0.92 a	37.19 ± 0.42 b	16.64 ± 1.41 a
	Ax1a	11.67 ± 0.57 a	41.50 ± 0.41 c	35.58 ± 0.91 c	13.62 ± 0.39 c
	Bx7a	8.33 ± 0.57 c	47.77 ± 1.34 b	34.37 ± 0.54 d	13.62 ± 0.98 c
	By8a	8.67 ± 0.57 c	41.50 ± 1.14 c	40.62 ± 0.4 2 a	12.00 ± 0.69 d
	Dx2a	10.67 ± 0.57 b	46.30 ± 0.32 b	36.48 ± 0.38 bc	15.89 ± 0.41 ab
	Dy12a	12.00 ± 1.00 a	40.93 ± 0.04 c	36.92 ± 0.16 b	15.28 ± 0.19 b
N120	WT	17.67 ± 0.57 a	54.43 ± 0.73 a	39.36 ± 1.23 a	24.80 ± 0.06 a
	Ax1a	17.00 ± 0.00 ab	47.20 ± 0.16 c	38.56 ± 0.98 ab	20.28 ± 0.33 c
	Bx7a	13.67 ± 0.57 c	50.20 ± 0.16 b	35.31 ± 0.52 c	20.39 ± 1.46 c
	By8a	12.33 ± 0.57 d	49.00 ± 0.48 b	39.71 ± 1.31 a	16.94 ± 0.11 d
	Dx2a	16.33 ± 0.57 b	47.13 ± 1.45 c	37.60 ± 0.35 b	23.13 ± 0.71 b
	Dy12a	16.67 ± 0.57 ab	47.00 ± 0.29 c	38.5 ± 0.79 ab	24.19 ± 0.12 ab
N180	WT	18.33 ± 1.15 a	54.07 ± 0.53 a	39.26 ± 0.21 ab	31.85 ± 0.77 a
	Ax1a	18.33 ± 0.57 a	48.40 ± 0.81 c	34.04 ± 0.83 c	21.35 ± 0.67 d
	Bx7a	17.00 ± 1.00 a	47.10 ± 0.81 cd	37.87 ± 1.06 b	23.58 ± 0.87 c
	By8a	13.33 ± 0.57 b	45.80 ± 0.41 d	39.53 ± 1.24 a	19.92 ± 0.32 d
	Dx2a	17.00 ± 0.00 a	49.80 ± 0.16 b	32.28 ± 0.83 d	25.59 ± 1.58 b
	Dy12a	17.67 ± 0.57 a	46.03 ± 0.74 d	39.16 ± 0.03 ab	26.64 ± 0.34 b
F-Value	N	495.378 **	132.842 **	14.454 **	907.557 **
	C	67.058 **	95.348 **	46.778 **	138.196 **
	N×C	4.14 **	10.88 **	15.561 **	12.274 **

Note: Data represent the means based on two years. Different letters in the same N level for different wheat lines indicate a significant difference (*p* < 0.05). ** indicate significance at the level of 0.01.

**Table 2 ijms-23-02178-t002:** Content of free sulfhydryl group (SH) and GMP, rheological characteristics of GMP gels in wheat lines with the absence of HMW-GS under different N treatments in 2017–2018 wheat growing season.

Treatment	Wheat Lines	Free SH Content (μmol/g)	GMP Properties				
			GMP Content (%)	Wet GMP-Gel Content (g/g)	G′ (Pa)	G″ (Pa)	δ (°)
N0	WT	0.88 ± 0.01 a	2.31 ± 0.18 a	0.98 ± 0.03 ab	2.23 ± 0.31 bc	0.97 ± 0.06 c	23.85 ± 3.39 b
	Ax1a	0.83 ± 0.02 b	1.96 ± 0.16 b	0.9 ± 0.02 b	2.01 ± 0.56 c	1.24 ± 0.13 b	32.69 ± 5.68 a
	Bx7a	0.82 ± 0.02 b	1.98 ± 0.13 b	0.94 ± 0.02 ab	2.09 ± 0.04 bc	1.01 ± 0.08 c	25.53 ± 1.54 b
	By8a	0.85 ± 0.01 ab	2.23 ± 0.11 a	1.08 ± 0.08 a	2.61 ± 0.44 b	1.23 ± 0.14 b	25.59 ± 4.53 b
	Dx2a	0.78 ± 0.01 c	2.09 ± 0.20 ab	0.95 ± 0.18 ab	1.73 ± 0.11 c	0.78 ± 0.01 d	24.33 ± 1.55 b
	Dy12a	0.82 ± 0.01 b	2.29 ± 0.08 a	0.95 ± 0.03 ab	3.68 ± 0.07 a	1.57 ± 0.08 a	23.21 ± 0.89 b
N120	WT	1.20 ± 0.01 a	3.01 ± 0.20 a	1.17 ± 0.11 ab	4.13 ± 0.11 bc	1.29 ± 0.11 d	17.35 ± 1.04 c
	Ax1a	1.20 ± 0.02 a	2.24 ± 0.21 c	0.97 ± 0.09 c	3.69 ± 0.26 d	1.53 ± 0.03 bc	22.68 ± 1.15 a
	Bx7a	1.20 ± 0.04 a	2.29 ± 0.19 c	1.01 ± 0.02 c	3.87 ± 0.31 cd	1.38 ± 0.17 cd	19.61 ± 0.85 b
	By8a	1.22 ± 0.02 a	2.98 ± 0.14 a	1.3 ± 0.11 a	4.37 ± 0.13 b	1.64 ± 0.11 b	20.59 ± 0.83 b
	Dx2a	1.21 ± 0.02 a	2.43 ± 0.08 bc	1.02 ± 0.02 bc	2.81 ± 0.19 e	1.05 ± 0.08 e	20.51 ± 0.64 b
	Dy12a	1.20 ± 0.02 a	2.65 ± 0.15 b	1.11 ± 0.04 bc	6.87 ± 0.42 a	2.12 ± 0.19 a	17.23 ± 2.31 c
N180	WT	1.45 ± 0.03 a	3.15 ± 0.04 a	1.49 ± 0.16 a	5.16 ± 0.49 bc	1.32 ± 0.07 c	14.41 ± 0.72 c
	Ax1a	1.35 ± 0.01 c	2.35 ± 0.18 c	1.23 ± 0.11 b	4.64 ± 0.36 d	1.63 ± 0.33 b	19.27 ± 3.18 a
	Bx7a	1.38 ± 0.05 bc	2.45 ± 0.13 bc	1.35 ± 0.07 ab	4.83 ± 1.03 cd	1.41 ± 0.39 c	16.1 ± 2.53 b
	By8a	1.41 ± 0.02 ab	3.14 ± 0.05 a	1.42 ± 0.08 ab	5.44 ± 0.22 b	1.71 ± 0.07 b	17.41 ± 0.51 b
	Dx2a	1.37 ± 0.02 bc	2.60 ± 0.08 bc	1.37 ± 0.07 ab	3.56 ± 0.16 e	1.08 ± 0.06 d	16.92 ± 0.62 b
	Dy12a	1.42 ± 0.03 ab	2.70 ± 0.20 b	1.38 ± 0.09 ab	8.57 ± 0.95 a	2.19 ± 0.06 a	14.46 ± 1.27 c

Note: Different letters in the same N level for different wheat lines indicate a significant difference (*p* < 0.05).

## Data Availability

The data used to support the findings of this study are available from the corresponding author upon request.

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
