# Peer review of "Effects of Nitrogen Fertilizer on Quality Characteristics of Wheat with the Absence of Different Individual High-Molecular-Weight Glutenin Subunits (HMW-GSs)"

_ijms, 2022, doi:10.3390/ijms23042178_

Round 1

Reviewer 1 Report

Line 333: what is a semi-field experiment? Please clarify where the experiment took place and what was the experimental design.  How many pots were prepared? How many plants per pot were considered.

Line 333-335: What were the average temperatures and relative humidity during the performance of the experiments?

Line 361: Is there any difference between the micro-Kjeldahl method and the micro-Kjeldahl AACC (2000) that it is mentioned in line 365.  If yes, the provide reference for former.

Line 380: how many biological and technical replicates were considered for the gene expression experiment?

Line 385: What is the ADP-RF reference gene? Provide reference and GenBank ID

Line  385-390: Provide GenBank IDs for all the listed genes.

Table 1: How do you calculate the grain yield per pot?

Figure 2: what are the expected sizes of the selected protein subunits? It seems that everything is above or close to 100kDa, but according to lines 34-35 the size seems to be smaller.

Author Response

Dear Reviewer,

Thank you very much for giving us an opportunity to revise our manuscript, we also appreciate your valuable comments on our manuscript. Those comments are very helpful for revising and improving our paper. We have revised the manuscript carefully to address your comments. Revised parts are marked in red in the paper. The main corrections in the paper and the responses to the comments are as following:

Point 1: Line 333: what is a semi-field experiment? Please clarify where the experiment took place and what was the experimental design.  How many pots were prepared? How many plants per pot were considered.

Response 1: Thanks for your nice comments. The “semi-field experiment” is an experiment carried out in the field shed with plastic to prevent water during wheat growing stage. The semi-field experiment which was conducted in Tangquan Experiment Station of Nanjing Agriculture University, Nanjing (118°27' E, 32°05' N), Jiangsu Province, P.R. China (line 333-335). Six wheat lines (5 deletion lines and 1 WT) were planted in pots with three N levels, including 0 kg N ha−1 (N0, 0 g N/pot), 120 kg N ha−1 (N120, 0.629 g N/pot) and 180 kg N ha−1 (N180, 0.943 g N/pot). The experiment was randomly designed with three replicates. Overall, 540 pots were prepared at each growing year (line 346-350). Each pot was planted 21 wheat seeds, then thinned to 7 seedlings at third leaf stage (line 351-352).

Point 2: Line 333-335: What were the average temperatures and relative humidity during the performance of the experiments?

Response 2: Thanks for your nice comments. We have added the information of average temperatures in Table S1, and described it in line 335-338.

Point 3: Line 361: Is there any difference between the micro-Kjeldahl method and the micro-Kjeldahl AACC (2000) that it is mentioned in line 365.  If yes, the provide reference for former.

Response 3: Thanks for your nice advices. There is no difference between them. Now we revised the citation of micro-Kjeldahl method AACC (2000) in lines 366 to make it clear.

Point 4: Line 380: how many biological and technical replicates were considered for the gene expression experiment?

Response 4: Thanks for your nice comments. We sampled the wheat grains at 20 DAA at each pot (The experiment was randomly designed with three biological replicates, line 348-349) for the determination of gene expression, and three technical replicates were performed. Now we added the detailed information of determination of gene expression experiment in line 397-398.

Point 5: Line 385: What is the ADP-RF reference gene? Provide reference and GenBank ID.

Response 5: Thanks for your nice comments. ADP-RF is ADP-ribosylation factor, which was used as reference gene to calculate the relative gene expression. Now, we added the reference about ADP-RF and its GenBank ID in line 390-391.

Point 6: Line 385-390: Provide GenBank IDs for all the listed genes.

Response 6: Thanks for your nice comments, we have added the information of GenBank IDs for all the listed genes in line 391-396.

Point 7: Table 1: How do you calculate the grain yield per pot?

Response 7: At mature stage, grains of all plants in one pot were harvested and weighed to get yield data. So the result was a little different from multiplication of yield components.

Point 8: Figure 2: what are the expected sizes of the selected protein subunits? It seems that everything is above or close to 100kDa, but according to lines 34-35 the size seems to be smaller.

Response 8: Thanks for your nice comments, the molecular weight of HMW-GS is ranged 65-90+ KDa, we have revised it in line 34 and made it more accurate.

Other changes in the manuscript were also marked in red color.

We tried our best to improve the manuscript and hope that the correction will meet with approval.

Once again, thank you very much for your comments and suggestions.

Yours sincerely,

Qin Zhou

Reviewer 2 Report

The manuscript presents a study of nitrogen effect on quality characteristics of wheat with the absence of different HMW-GSs.

This document if presented with good quality, but needs to be improved

Please give the following information or restructure it:

  • In figure 1 you show gluten and protein components in different graphics, but you don’t indicate which component is. May be you can do the same of figure 3, where identify A-1 and A-2. I suppose that wet gluten (line 110) correspond to B1, dry gluten (line 113) correspond to B2 and the gluten index (line 114) correspond to B3, but I’m not sure and difficult to see the differences between samples. Moreover, I have the same problem with the protein components (C), in line 117 and 118 you specify which protein determined (glutenin, albumin, globulin and gliadin, but don’t identify the graphic where you can found the information.
  • There are different acronyms (such as DAA, NR, GS, GMP, among others) that are used in the document and their meaning is not known when you described your resuls. I think it would be interesting to add their meaning when it appears in the text the first time.
  • In line 166 you indicates that Dy12a do not show a significantly decrease of free SH content compared with other mutants (except By8a). The values presented in table 2 show the same value for Dy12a and Bx7a, and you consider than Bx7a decrease.

Author Response

Dear Reviewer,

Thank you very much for giving us an opportunity to revise our manuscript, we also appreciate your valuable comments on our manuscript. Those comments are very helpful for revising and improving our paper. We have revised the manuscript carefully to address your comments. Revised parts are marked in red in the paper. The main corrections in the paper and the responses to the comments are as following:

This document if presented with good quality, but needs to be improved

Please give the following information or restructure it:

Point 1: In figure 1 you show gluten and protein components in different graphics, but you don’t indicate which component is. May be you can do the same of figure 3, where identify A-1 and A-2. I suppose that wet gluten (line 110) correspond to B1, dry gluten (line 113) correspond to B2 and the gluten index (line 114) correspond to B3, but I’m not sure and difficult to see the differences between samples. Moreover, I have the same problem with the protein components (C), in line 117 and 118 you specify which protein determined (glutenin, albumin, globulin and gliadin, but don’t identify the graphic where you can found the information.

Response 1: Thanks for your nice comments. We have revised title of Figure 1 to indicate which components is in line 124-125 and text of section 2.2 to make it clear.

Point 2: There are different acronyms (such as DAA, NR, GS, GMP, among others) that are used in the document and their meaning is not known when you described your resuls. I think it would be interesting to add their meaning when it appears in the text the first time.

Response 2: Thanks for your nice comments. We have added the full name of acronym when it appeared in the text at first time, and the revised information are shown in lines 21, 162, 196-197 210.

Point 3: In line 166 you indicates that Dy12a do not show a significantly decrease of free SH content compared with other mutants (except By8a). The values presented in table 2 show the same value for Dy12a and Bx7a, and you consider than Bx7a decrease.

Response 3: Thanks for your nice comments. We have noticed the descriptive questions for free SH, we have revised it in line 165-169.

Other changes in the manuscript were also marked in red color.

We tried our best to improve the manuscript and hope that the correction will meet with approval.

Once again, thank you very much for your comments and suggestions.

Yours sincerely,

Qin Zhou